# "Shutdown" of the Proton Exchange Channel Waveguide in the Phase Modulator under the Influence of the Pyroelectric Effect

**Roman Sergeevitch Ponomarev** [1,2,*]**, Denis Igorevitch Shevtsov** [1] **and Pavel Victorovitch Karnaushkin** [1,2]

[1]   Perm State National Research University, Solid State Physics Department, 614990 Perm, Russia; shevtsov@pnppk.ru (S.D.I.); pavelkarn2@gmail.com (K.P.V.)

[2]   Perm Federal Research Center, Ural Branch of Russian Academy of Sciences, 614990 Perm, Russia

[*]   Correspondence: rsponomarev@gmail.com; Tel.: +7-922-315-1003

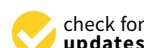

**Featured Application: The results can have application in fiber optic gyroscopes to avoid the breakdown during the fast heating of device.**

**Abstract:** It is shown that the termination of the channeling of the fundamental radiation mode in the waveguide can be observed upon heating of an optical integrated circuit based on proton exchange channel waveguides formed in a lithium niobate single crystal. This process is reversible, but restoration of waveguide performance takes tens of minutes. The effect of the waveguide disappearance is observed upon rapid heating (5 K/min) from a low temperature (minus 40 °C). This effect can lead to a temporary failure of navigation systems using fiber optic gyroscopes with modulators based on a lithium niobate crystal.

**Keywords:** channel waveguide; proton exchange; fundamental mode; pyroelectric effect; mobile charge; fiber optic gyroscope; phase modulator

## 1. Introduction

Optical integrated circuits based on a lithium niobate crystal are used in fiber optic gyroscopes for phase modulation of a signal and in fiber trunk circuits for signal encoding at a frequency of up to 100 GHz [1,2]. The main advantages of using a lithium niobate crystal are its wide transmission window, high electro-optical coefficient $r_{33}$ and the relative simplicity of creating waveguides [3]. The disadvantages of the crystal include its complex structure and the variety of defects, as well as the difference in the composition, structure and properties of the near-surface layers on crystals of different manufacturers [4]. The waveguides of optical integrated circuits are formed by the ion exchange method. As a rule, they either replace $Li^+$ ions with $H^+$ ions (proton exchange method), or $Nb^{5+}$ ions with $Ti^{5+}$ ions (titanium diffusion method). The proton exchange method is somewhat simpler, because it does not require heating the crystal above 350 °C [5].

In this paper we studied the temperature behavior of phase signal modulators used in a fiber optic gyroscope and constructed according to the scheme of Y-splitter. Temperature effects limit the accuracy of fiber optic gyroscopes and affect their noise characteristics [6–8]. Phenomena of a similar nature are observed in radiation intensity modulators of optic fiber transmission system [9,10].

A model, in which charged defects in the matrix of lithium niobate can move near the waveguide, was proposed to explain the observed drifts. In this model, the refractive index of the waveguide changes due to the local electro-optical effect.

The output power was used as the measurement signal, and the variable external temperature in the temperature chamber was used as the effect.

## 2. Experimental Samples

The radiation phase modulators constructed according to the scheme of a Y-splitter with a division factor of 1/1 (Figure 1) were used as samples for research.

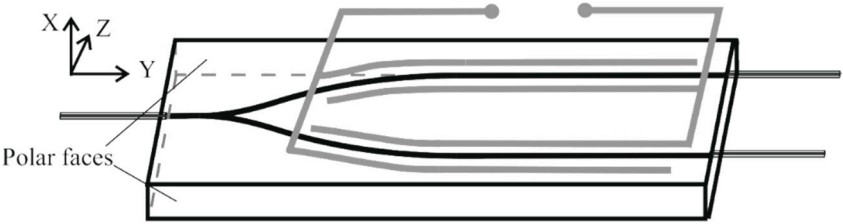

**Figure 1.** Topology of the waveguides of the phase modulator. Black lines are optical waveguides, grey lines are surface electrode system.

Optical channel waveguides were formed by the proton exchange method [11] in benzoic acid (2 h, 175 °C), followed by annealing (4 h, 350 °C, air atmosphere). Waveguide width was 7.5 um, $\Delta n \approx 0.01$. The process was selected in such a way that the shape of the propagating beam and the numerical aperture of the waveguide maximally corresponded to the shape and aperture of the core of Panda-type optical fiber used to input and output radiation in this type of modulator. A lithium niobate of congruent composition (X-cut) was used as the virgin crystal. In this case, the waveguides were propagating along the *Y* axis, and the pyroelectric field corresponded to Z-direction. The length of the sample was 38 mm, 22 mm of which were in the straight-line portions of the waveguides. The sample width $d = 3.2$ mm. The fiber-to-fiber optical losses was 6 dB.

All samples had surface Cr–Au electrode system without special buffer layers. Electrodes were not connected to any wires and did not short circuit during the experiments.

We used clean samples to study how the pyroelectric effect influences output optical signals. To make sure the pyroelectricity is the reason of signal changing we used samples covered by conductive graphite paste. It was applied on the side faces and on the lower surface of the samples in order to bring about closure of the polar faces of the crystal and the rapid relaxation of the pyroelectric surface charges.

## 3. Experimental Technique

The programmed precision temperature chamber Espec MC-711 was used to study the action of temperature on the output optical power. The temperature of the sample during the exposure was considered equal to the temperature of the air in the chamber, which was controlled using the embedded sensor. A sample in an open process container was placed in a heat chamber in the upper part of the working volume. The modulator chip was placed on an aluminum substrate in order to reduce the temperature gradient in the sample. The chip and the substrate were connected using a thin layer of silicone-based adhesive with high thermal conductivity.

The experimental assembly is shown in Figure 2.

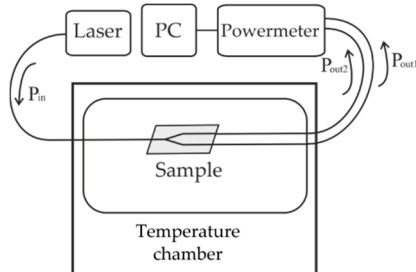

**Figure 2.** Scheme of the experimental assembly for optical measurements.

The sample was connected to the radiation source and receiver by welding fiber optic light guides. The supply and return fiber optic light guides were brought out of the temperature chamber. The construction of the chamber door excluded the bending and squeezing of fiber optic light guides during the experiment due to soft gaskets. The radiation source and receiver were located outside the chamber and were not exposed to variable temperature. A fiber laser with an output power of 5 mW and a central wavelength of 1550 nm was used as a source of radiation.

The measurements were carried out both in constant temperature regime and with a change of temperature. During measurements, the time dependence of beam output power $P_{out}$ was recorded by Santec PEM 330 optical power meters (Japan).

## 4. Experimental Results

The results of measurement $P_{out}(t)$ for Y-splitter during temperature cycling of the sample and the action of the pyroelectric effect are presented in Figures 3 and 4, where the red and black curves correspond to the two arms of the Y-splitter, and the gray broken line corresponds to the temperature in the temperature chamber during the experiment. The range of temperature changes was 140 °C. Because of the long thermocycle we showed only the most important part of signal changing and *X*-axis start not from zero.

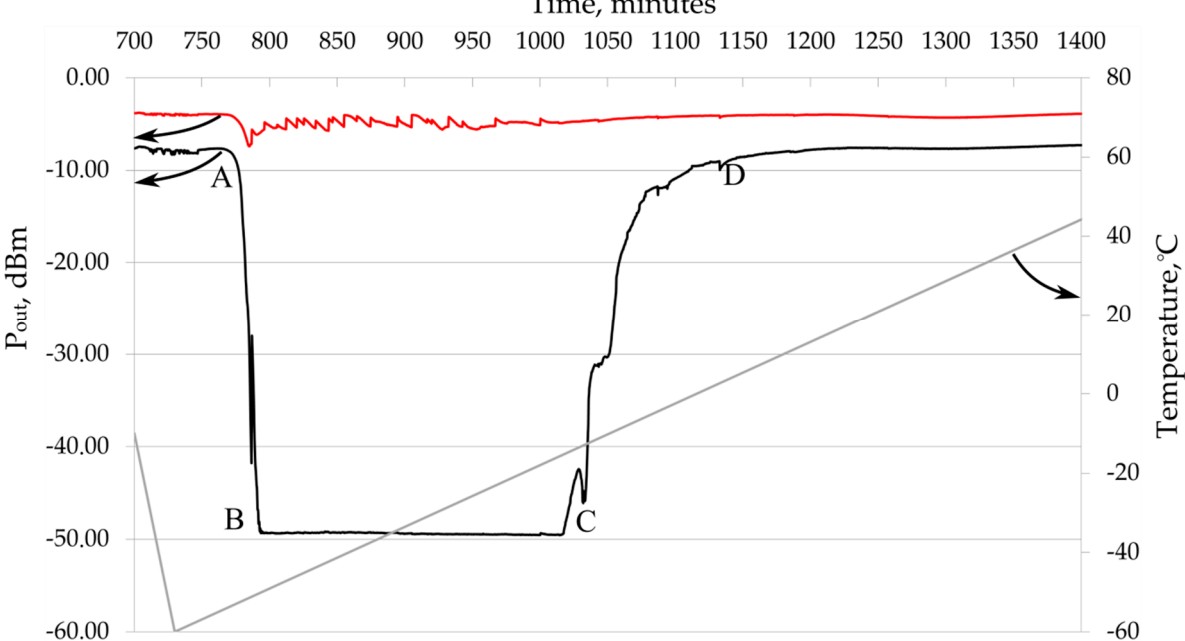

**Figure 3.** The change in the output signal of the Y-splitter under the action of the pyroelectric effect. Black and red lines are $P_{out1}$ and $P_{out2}$ respectively, gray line is temperature. The rate of temperature increase is 0.16 °C/min, the lowest sample temperature is about −55 °C.

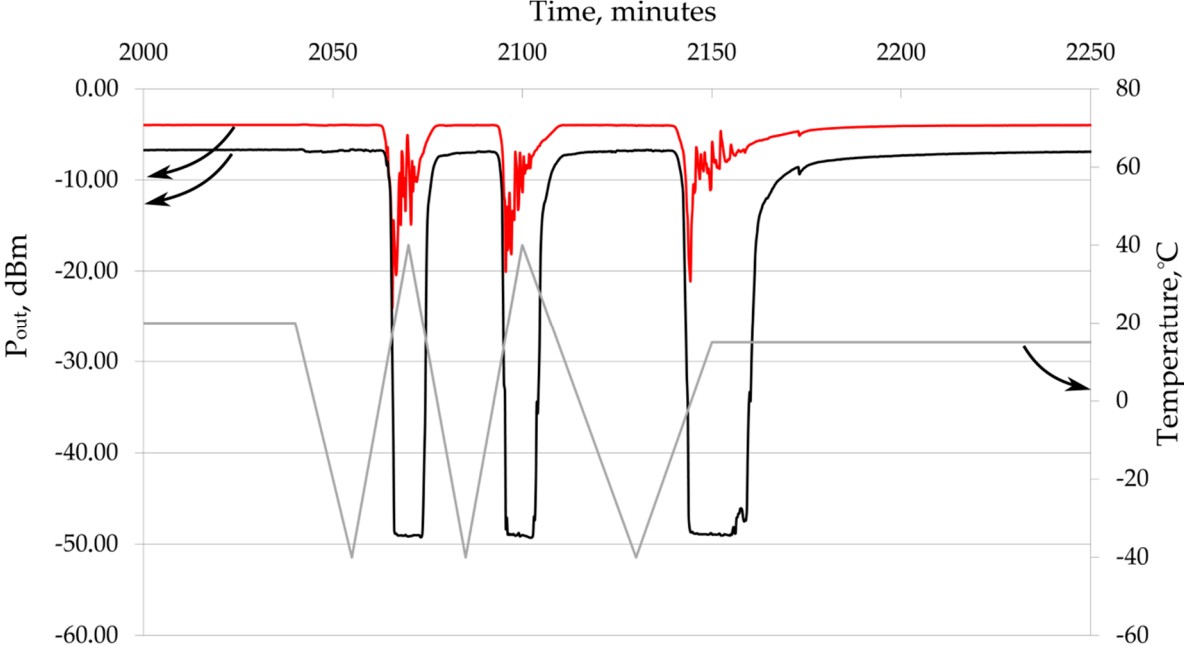

**Figure 4.** $P_{out}$ degradation upon heating at a rate of 5.3 and 2.5 °C/min. Black and red lines are $P_{out1}$ and $P_{out2}$ respectively, gray line is temperature.

As can be seen from the presented figure, a sharp drop in the output optical power (~40 dBm) is observed in the heating areas on one arm of the Y-splitter. During subsequent heating, the $P_{out}$ value is restored, and this behavior is typical for all heating cycles. The cooling of the sample does not give a similar effect, and the analysis of the graph does not allow us to identify a stable temperature at which $P_{out}$ value decreases.

The power decrease for such a heating rate is characteristic of only one arm of the Y-splitter. At the same time, a small power output variation is observed in the second arm. The considered power drop can be conditionally classified into three sections: A–B is a sharp increase in optical loss ("disappearance" of the waveguide), B–C is the absence of the waveguide and C–D is the waveguide restoration. The length of the section *A-B* in the presented figure is 24 min, the length of B–C is 5.5 h and the length of C–D is about 2.5 h. The signal power in the section B–C (−49 dBm) corresponds to the value that would be observed in the absence of a waveguide channel, but in the presence of input radiation.

For higher heating rates, similar phenomena are observed, accompanied by changes in the signal of the second arm of the Y-splitter (Figure 4). The heating rate is 5.3 and 2.5 K/min.

The condition for the waveguide "disappearance" was also investigated from the point of view of the starting temperature used for sample heating. For this purpose, a thermal cycle was proposed in which the sample was heated at different speeds under the influence of temperatures of minus 40 °C, minus 20 °C and 0 °C. The heating rates varied during the thermal cycle, but the temperature drop remained constant and was equal to 60 °C. The results show that the depth of $P_{out}$ degradation heavily depends on the temperature at which the sample was heated. Short-term, but complete "disappearance" of both waveguides is observed at an initial temperature of minus 40 °C. With an increase in the initial heating temperature, this phenomenon weakens and completely disappears when the sample is heated at the temperature of 0 °C. For this reason, the phenomena we discovered may not have been observed previously by other researchers, since the used parameters of the thermal cycle (low initial temperature, high heating rate and large temperature differential) are not typical for testing of optical integrated circuits.

The sample covered by conductive paste was measured to make sure that $P_{out}$ changes are due to pyroelectric effect. Signal changes are about 1 dBm and optical connections have good stability (Figure 5).

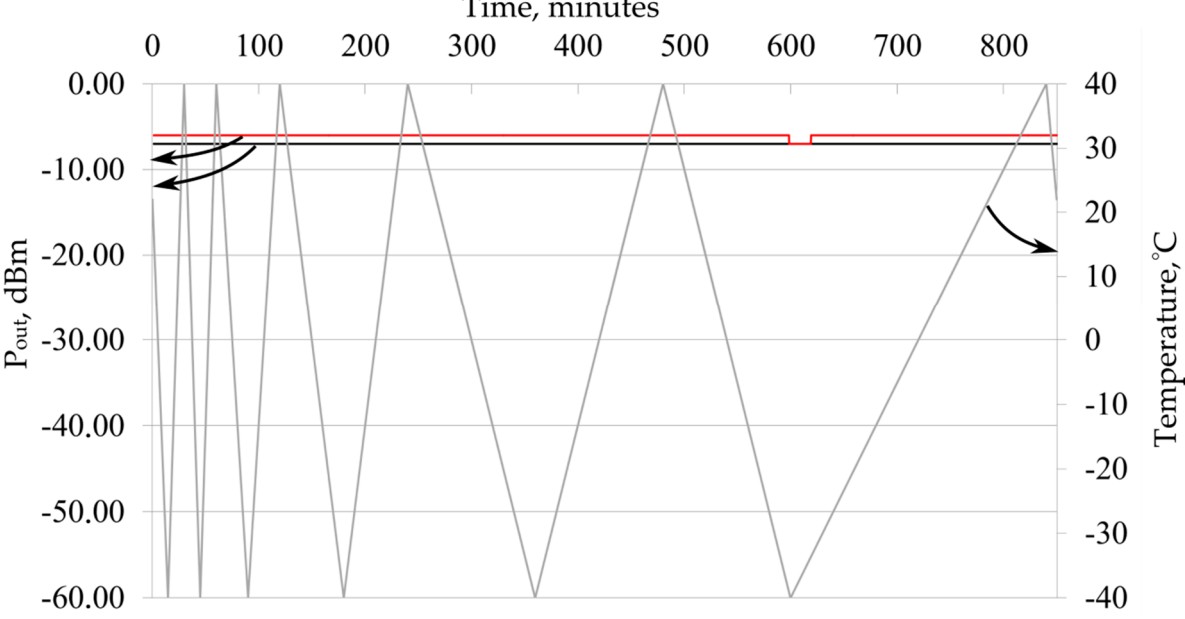

**Figure 5.** $P_{out}$ and temperature plots versus time for sample covered by conductive paste. Black and red lines are $P_{out1}$ and $P_{out2}$ respectively, gray line is temperature.

## 5. Discussion of Measurement Results: Model of Charged Defects Motion Near a Waveguide

The following facts were established during temperature tests on a large number of samples of Y-splitters with an active pyroelectric effect:

1. The disappearance of the waveguide is observed only when the sample is heated.
2. The output power varies differently on the two arms of the modulator.
3. These phenomena disappear when the edges of the sample are closed with a conductive paste.
4. The observed power output does not correspond to a complete laser shutdown, but to the input of radiation into the crystal without a waveguide.
5. In some samples, accompanying phenomena were observed without the disappearance of the channel. These samples differed from other samples by the width of their waveguides and the mode of their creation.
6. When the temperature is stabilized, the power in the arms of the Y-splitter is restored to the initial values.

The given pieces of evidence are interpreted from the point of view of the model of a channel waveguide surrounded by mobile charged defects. Let us consider experimental facts in detail.

### 5.1. The Disappearance of the Waveguide Is Observed only When the Sample Is Heated

We offer a simple model of how the pyroelectric effect can influence the waveguide refractive index. During waveguide formation, mobile $H^+$-ions are indiffused in the crystal near-surface layer and form some new phases with different lattice parameter [12]. Dislocations need to arise during this process on the waveguide border to prevent crystal destruction because of lattice parameter mismatch. A dislocation grid formed on oxygen ion-lattice in $LiNbO_3$ and has negative charge. It attracts $H^+$-ions and they are placed around the waveguide and can move along dislocation lines like on tubes. Thus waveguide is partially shielded from external electric field and also pyroelectric

field by the surrounding ions which decrease electric field intensity inside the waveguide. In this way waveguide and bulk crystal will have a difference in refractive index changing during action of pyroelectric effect.

This effect works only for sample heating because refractive index of bulk crystal is increased when the pyroelectric field is opposite to the spontaneous polarization direction and waveguide contrast $\Delta n$ becomes lower. If we cool the sample waveguide contrast $\Delta n$ becomes higher and waveguide "moves away" from danger condition.

### 5.2. In the Study of Y-splitters, It Was Found That Only One Arm of the Splitter Can Be Affected by This Phenomenon (Or One Arm Can Be Affected Much More Than the Other)

Also, as a system with mobile charges, the arms of a Y-splitter are a coupled system because they have a junction point. In this case, the mobile charges, which are uniformly distributed near the two arms of the Y-splitter in the absence of the pyroeffect, can be redistributed under the influence of the pyroeffect so that their number near one arm becomes larger than near the other arm. Then the effect of their impact will be different for the two arms. An approximate diagram of such a process is shown in Figure 6.

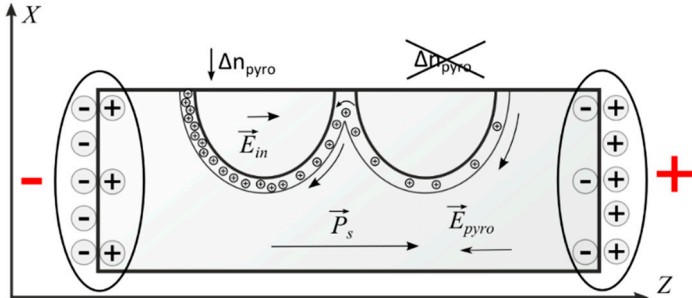

**Figure 6.** Charge flow under the influence of a pyroelectric field in the region of waveguides connection.

This phenomenon leads to the fact that one arm disappears completely, and the second arm does not disappear at all or partially disappears for a short time.

### 5.3. These Phenomena Disappear When the Edges of the Sample Are Closed With a Conductive Paste

When the edges of the crystal are closed, the relaxation of pyroelectric charges occurs in a short time. The strength of the pyroelectric field does not reach values sufficient for the disappearance of the waveguide.

### 5.4. The Observed Power Output Does Not Correspond to a Complete Laser Shutdown, but to the Input of Radiation into the Crystal without a Waveguide

When the waveguide disappears, the radiation entering the crystal from an optical fiber connected to the laser does not disappear, but propagates freely throughout the crystal. A small part of this radiation falls on the end of the output optical fiber and is recorded by the optical power meter.

### 5.5. In Some Samples, Accompanying Phenomena Were Observed without the Disappearance of the Channel. These Samples Differed from Other Samples by the Lithium Niobate Wafers Manufacturer

For some samples, the channels did not disappear, which may be linked to a different actual width and depth of the waveguide, as well as its contrast ratio. These parameters directly depend on the resulting refractive index distribution and near-waveguide charge mobility. These parameters strongly depend on the wafer's near-surface layer condition like dislocation density or hydrogen ions diffusion coefficient by depth. However, these wafer parameters are not specified. It should be noted

that phenomena indicating an uneven refractivity variation in two waveguides were observed on all samples.

### 5.6. When the Temperature is Stabilized, Waveguide Properties Are Restored over Time

When the temperature is stabilized, mobile charges existing within the crystal and on its surface (as well as charged particles attracted by the electric field from the air surrounding the crystal) shield the pyroelectric charges, leading to a decrease in the pyroelectric field in the crystal. Thus, after some time and at a constant temperature, the pyroelectric field drops to a value at which the waveguide contrast sufficient for operation in the single mode is restored.

## 6. Theoretical Interpretation

The disappearance of the waveguide is observed only when the sample is heated.

An increase in the refractive index throughout the entire crystal volume occurs with an increase in the crystal temperature and leads to the waveguide contrast $\Delta n = n_2 - n_1$ compression, as well as to the termination of the channeling of the fundamental radiation mode (when the limiting contrast value is reached). The critical value $\Delta n_{crit}$ can be estimated as follows:

For simplicity, let us consider the propagation of a transverse-electric wave (TE-wave) with a wavelength $\lambda$ along a planar waveguide with a step-graded index (Figure 7).

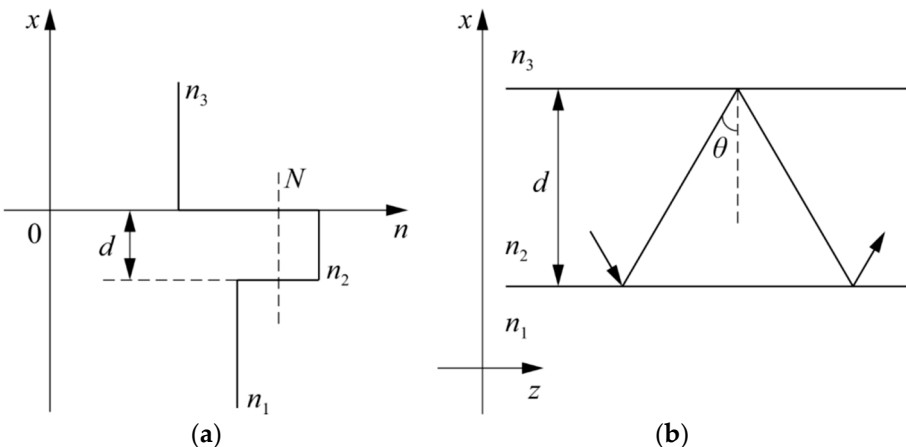

**Figure 7.** A waveguide model with a step-graded index: (**a**) a refractive index profile in a planar waveguide; (**b**) the scheme of light propagation within a waveguide in the ray approximation. $n_1 = 2.20$, $n_2 = 2.21$, $n_3 = 1$ are the refractive indices of the substrate, waveguide and coating (respectively), $N$ is the effective refractive index, $d$ is the thickness of the waveguide and $\theta$ is the angle of incidence of the beam on the interface.

The light propagation in the waveguide is described by single equation for a TE-wave (1):

$$\frac{\partial^2 E_y(x)}{\partial x^2} + \left(\frac{n^2\omega^2}{c^2} - \beta^2\right)E_y(x) = 0, \tag{1}$$

For the region $x > 0$ in Figure 7a, the solution to Equation (1) is:

$$E_y(x) = E_3 e^{-\gamma_3 x}, \tag{2}$$

where $\gamma_3 = \frac{\omega}{c}\sqrt{N^2 - n_3^2}$.

For the region $-d < x < 0$ in Figure 7a, the solution to Equation (1) is:

$$E_y(x) = E_2 \cos(k_x x + \varphi_2), \tag{3}$$

where $k_x = \frac{\omega}{c}\sqrt{n_2^2 - N^2}$.

For the region $x < -d$ in Figure 7a, the solution to Equation (1) is:

$$E_y(x) = E_1 e^{-\gamma_1(x+d)}, \tag{4}$$

where $\gamma_1 = \frac{\omega}{c}\sqrt{N^2 - n_1^2}$.

Based on the boundary conditions on the media interfaces $n_3$ and $n_2$, $n_1$ and $n_2$:

$$\begin{aligned} E_3 &= E_2\cos(\varphi_3) & E_1 &= E_2\cos(-k_x d + \varphi_3) \\ \gamma_3 E_3 &= k_x E_2\sin(\varphi_3) & \gamma_1 E_1 &= -k_x E_2\sin(-k_x d + \varphi_3) \end{aligned}, \tag{5}$$

By dividing the lower equation from Equation (5) by the upper equation, we obtain the following system:

$$\begin{aligned} \tan\varphi_3 &= \frac{\gamma_3}{k_x} \\ \tan(k_x d - \varphi_3) &= \frac{\gamma_1}{k_x} \end{aligned}, \tag{6}$$

If we express the arguments of tangents in Equation (6) and substitute the upper equation into the lower one, we obtain the dispersion equation:

$$k_x d = \tan^{-1}\left(\frac{\gamma_1}{k_x}\right) + \tan^{-1}\left(\frac{\gamma_3}{k_x}\right) + m\pi, \tag{7}$$

By substituting $k_x$, $\gamma_1$ and $\gamma_1$ in Equation (7), we obtain the following equations:

$$\frac{2\pi}{\lambda}d\sqrt{n_2^2 - N^2} = \tan^{-1}\left(\sqrt{\frac{N^2 - n_1^2}{n_2^2 - N^2}}\right) + \tan^{-1}\left(\sqrt{\frac{N^2 - n_3^2}{n_2^2 - N^2}}\right) + m\pi, \tag{8}$$

By making substitutions $V \equiv \frac{2\pi}{\lambda}d\sqrt{n_2^2 - n_1^2}$, $b_E \equiv \frac{N^2 - n_1^2}{n_2^2 - n_1^2}$, $a_E \equiv \frac{n_1^2 - n_3^2}{n_2^2 - n_1^2}$, we obtain the normalized dispersion equation:

$$V\sqrt{1 - b_E} = \tan^{-1}\left(\sqrt{\frac{b_E}{1 - b_E}}\right) + \tan^{-1}\left(\sqrt{\frac{a_E + b_E}{1 - b_E}}\right) + m\pi, \tag{9}$$

In the case of a symmetric waveguide ($n_1 = n_3$, $a_E = 0$), Equation (9) always has a solution, at least for the value $m = 0$. In the case of an asymmetric waveguide ($n_1 \neq n_3$, $a_E \neq 0$), not only the solutions for positive $m$, but also the solutions for $m = 0$ successively disappear with decreasing $\Delta n = n_2 - n_3$ at a fixed layer thickness $d$. Physically, this phenomenon can be explained by the fact that the fundamental mode begins to radiate into the substrate. The cutoff condition for the fundamental mode in an asymmetric waveguide is as follows:

$$V_0 = \tan^{-1}(\sqrt{a_E}), \tag{10}$$

The following equation was graphically solved in order to find $\Delta n$:

$$\frac{2\pi}{\lambda}d\sqrt{n_2^2 - n_1^2} = \tan^{-1}\left(\sqrt{\frac{n_1^2 - n_3^2}{n_2^2 - n_1^2}}\right), \tag{11}$$

where $\lambda = 1.55\ \mu m$; $d = 7\ \mu m$; $n_1$ from 2.17 to 2.22; $n_2 = 2.21$, $n_3 = 1$.

The required value of $\Delta n_{crit}$ was 0.00085.

## 7. Calculation of the Pyroelectric Effect for a Waveguide in Lithium Niobate

We will calculate the electric voltage arising on the faces of the Y-splitter under the influence of the pyroelectric effect when the temperature of the crystal changes by 1 K. In a first approximation, we will consider a Y-splitter as a flat capacitor, the capacitance of which is determined by the relation $C = \varepsilon_{33}\varepsilon_0 S/d$, where $\varepsilon_{33}$ is the relative permittivity in the Z-direction and $d$ is the width of the sample in the Z-direction. For lithium niobate, $\varepsilon_{33} = 30$ [13] for low frequency voltage applied. It should be noted that in the calculations of microwave devices and optical devices, $\varepsilon_{33}$ decreases to a value of about 5 units based on the value of the refractive index for lithium niobate. The product of $\varepsilon_{33}$ and the dielectric constant $\varepsilon_0$ gives the absolute dielectric constant of lithium niobate in the system of SI units, measured in *F/m*.

The charge that appears on the faces of the Y-splitter under the influence of the pyroelectric effect is calculated by the formula $Q = \gamma \cdot \Delta T \cdot S$, where $\Delta T$ is the change in temperature of the sample and $S$ is the area of the side face. Then the ratio $U = \gamma \cdot \Delta T \cdot d/\varepsilon_{33}\varepsilon_0$ is true for the voltage that arises on the faces of the Y-splitter. For $\Delta T = 1$ K, we will obtain the voltage value at the faces of the integrated optical circuit phase modulator $U = 5.5 \cdot 10^2 \ V/K$.

Thus, when the temperature of the Y-splitter changes to 1 K, a voltage of 550 V appears on its faces. In this case, the electric field strength $E$ in the crystal can reach 1760 V/cm. When the crystal temperature changes by 100 K, which occurs at standard thermal cycles, the electric field strength can reach $E = 1.76 \cdot 10^5$ V/cm and lead to change of waveguide refractive index.

The $\Delta n_{pyro}$ caused by this electric field is calculated by formula $\Delta n_{pyro} = -\frac{1}{2}r_{33}n_e^3 E \approx 0.003$ for $\Delta T = 100\ °C$. For waveguide "shutdown" $\Delta n = n_2 - n_1$ need to achieve $\Delta n_{crit} = 0.00085$. In this case decrease of waveguide refractive index need to be $n_2 - (n_1 + \Delta n_{crit}) = 0.00915$, but pyroelectric effect can give only 0.003. It has the same order of magnitude than $\Delta n_{crit}$, but 3 times lower than we need to have for waveguide "shutdown". Now we have no exact explanation for this effect, but it could be connected with electrodes deposited near to the waveguide and redistributing the charges near these electrodes. Also, the pyroelectric coefficient of near-surface layer could be higher because of the difference of composition and properties between surface layer and bulk crystal. Additionally, $\Delta n_{crit}$ calculations are very simple and do not take account of real refractive index distribution.

## 8. Conclusions

The effect of termination of the channeling of the fundamental radiation mode in proton exchange channel waveguides formed in a lithium niobate single crystal was experimentally demonstrated for the first time. "Dangerous" temperature conditions were determined and the critical value $\Delta n_{crit}$, at which the channeling of radiation ceases, was calculated. It is shown that the magnitude of the pyroelectric effect is not sufficient for achieving the specified critical value $\Delta n_{crit}$, which requires follow-up studies of physical mechanisms leading to the "shutdown" of the waveguide.

**Author Contributions:** P.R.S.—experiment and results analysis, S.D.I.—samples and equipment, K.P.V.—theory and calculations.

**Funding:** This research and APC was funded by RFBR and Perm Region according to the research project № 19-48-590018.

**Conflicts of Interest:** The authors declare no conflict of interest.

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
