# Peer review of "“Shutdown” of the Proton Exchange Channel Waveguide in the Phase Modulator under the Influence of the Pyroelectric Effect"

_applsci, doi:10.3390/app9214585_

Round 1

Reviewer 1 Report

This work presents observation of a phenomenon which could negatively impact the operation of fiber optic gyroscopes, namely the failure of one of the components under harsh temperature increases from low temperatures. This component, the lithium niobate y-splitter and phase modulator is a critical component in such systems and therefore this research is of significant interest to researchers and companies who work with fiber optic gyroscopes. This is an interesting result that must be presented in a more rigorous manner. If these results are presented with a theory that describes the results they are novel and significant. If the authors cannot present a cogent theoretical description the results are still novel but irrelevant theory should be removed from the manuscript.

My primary concern with this paper is the incomplete presentation of results to back-up the claims made, and analysis inconsistent with presented results. The primary claim made in this paper is that when a proton exchanged lithium niobate y-splitter and phase modulator is heated quickly from -40C a pyroelectric effect occurs which locally changes the refractive index of the proton exchanged waveguide in such a way that it no longer guides light.

Firstly, no conclusive evidence is presented that the increase in transmission loss is not due to a decoupling between the integrated device and the fibre connected to it. Such conclusive evidence can be reasonably presented by including the transmission vs temperature plots when the chip has conductive graphite paste applied.

Secondly, it is stated on page 4 line 24 and in section 5.4 that 'In some samples, accompanying phenomena were observed without the disappearance of the channel. These samples differed from other samples by the width of their waveguides and the mode of their creation.' This assertion is presented without accompanying data to show the change in transmission with heating for samples fabricated differently. This data should be presented and discussed in depth to quantitatively demonstrate any relationship between fabrication conditions and the 'shutdown' phenomenon. This information is critical to the impact of the paper as it can demonstrate fabrication conditions suitable for high reliability chips for gyroscopes.

Thirdly, the description of the model presented to describe the imbalance in the 'shutdown' between the two arms is too brief. The authors should state what is the origin of the mobile charge, and why it is trapped near the waveguides. The authors should also explicitly state how the distribution of charge affects the refractive index of both the waveguide and the bulk crystal. As a reader this model does not make it clear why there is a larger pyroelectric effect on one waveguide compared to the other. If charges are indeed trapped near the proton exchanged region, it is not clear why there is a larger pyroelectric field on one waveguide if the two waveguides 'have a junction point' i.e. are electrically connected.

Fourthly, the results presented in section 6 do not explain what they purport to explain in the title, i.e. 'the disappearance of the waveguide is observed only when the sample is heated'. Instead they simply calculate a refractive index change required to prohibit guiding. There is no discussion as to why the refractive index of the proton exchanged region would change via the pyroelectric effect without the refractive index of the substrate changing. The is also no discussion as to why the effect is not observed in the other arm in the cooling cycle.

Fifthly, the theoretical description is not close to matching the experimental results, and a discussion as to why the theory presented can be reasonably considered to account for some of the refractive index change is needed.

Finally some general comments and questions to improve the presentation.

On page 2 line 5, the fabrication parameters of the waveguide should be shown more explicitly, i.e. proton exchange depth, annealing time, effective refractive index, peak refractive index, waveguide width. A reference should also be added for the annealed proton exchange technique. For example ML Bortz and MM Fejer, Optics Letters 16, 23 (1991).

On page 2 line 9 X-section should be changed to the standard terminology 'x-cut'

On page 2 line 10, it is more standard to use the word propagating to describe the direction of the waveguide instead of the word located.

On page 2 line 14 it is not clear what is meant by polar faces where the graphite conductive paste is applied. It would be instructive to add this to figure 1.

One page 2 line 29, it would be instructive to elaborate on the process used to splice the fibre to the chip as it is clearly resistant to misalignment with fast temperature changes.

In Figure 1 it is not clear how and where external electric field is applied. If there are electrodes in this design they should be shown in Figure 1. If there is a buffer layer under the electrodes it should also be shown, and inlcuded in the analysis in section 6.

Figure 3 and 4 are poorly presented. Effort should be made that all labelling is clear and not covered by lines. It is not clear why the time axis does not start at zero. Labelling of each trace should be done in the more standard method of a legend or discussion in the caption.

The use of the symbols Iin and Iout to represent power is confusing as I is usually reserved in these contexts for current or intensity.

In section 6, the real values for this system should be used in the presented model. Additionally in section 6 the derivation of the wave equation is unnecessary  and is not related to the theme of this research.

On page 8 line 23, values in scientific papers should always be presented in SI units. As relative permittivity is unitless the specification of units here is meaningless.

On page 9 line 9, the correct unit is K (Kelvin) not degrees Kelvin, the text should also read 'changes by' rather than 'changes to'

On page 9 line 10, the width of the sample is not stated in the article, but the calculation implies it is 3.1mm wide, is this correct? Regardless the width of the sample is relevant to this calculation and should be stated.

On page 9 line 12 the word refraction should read refractive

Author Response

Dear Reviewer, thank You for your comments and questions. It was very useful to improve the manuscript and eliminate some logical mistakes and inconsistency.

Please see our responses in attachment.

Reviewer 2 Report

In this paper, the authors discovered the disappearance of the lithium niobate proton-exchanged waveguide upon heating. They discussed the experimental phenomena in detail and tried to explain them by the pyroelectric effect. Through some calculations, they found the magnitude of the pyroelectric effect is not sufficient to lead to the absence of the waveguide. As the work is incomplete and this defect can be simply avoided by using conductive paste, I don’t think this paper is suitable for Applied Sciences now.

Specifically:

Since the proper physical mechanisms have not been found, I suggest deleting “under the influence of the pyroelectric effect” in the paper title. For the part of “Experimental samples”, can you provide the data of the beam propagation loss and the fiber-waveguide coupling loss in the paper so that the reader can review the “shutdown” influence numerically? For the part of “Experimental results”, figure 3 was plotted very roughly. It was a little confusing for the vertical axis of temperature, and the corresponding relationship between the signal curves and the Y-splitter arms should be marked in the figure. For figure 4, were the experimental results exactly the same with the heating rate of 5.3 and 2.5 °C/min? In addition, the test results of the samples with conductive paste should be added for comparison. For the 5th part, can you explain the mobile charges model at length? What if there was a single waveguide channel? According to the experimental facts in some samples, different width and depth of the waveguide only resulted in accompanying phenomena without the disappearance of the channel, the boundary or the range of waveguide width and depth corresponds to the “shutdown” need to be mentioned to be published in any journal. For the 6th part, the final calculation result of the waveguide index contrast critical value should be checked again.

Author Response

Dear Reviewer, thank You for your comments and questions. It was very useful to improve the manuscript understandability.

Please see our responses in attachment.

Round 2

Reviewer 1 Report

The authors have sufficiently addressed the issues with the manuscript and it now presents a logical and consistent account of the observed phenomenon.

Author Response

Hello! Thank you for your comments! We add some more explanations in the text and correct some little mistakes.

Reviewer 2 Report

I am glad that the experimental part was presented better in the revised version. The paper is almost ready to be published in Applied Science. But there are still some minor problems to be rectified.

The newly added grey lines in figure 1 should be clarified in the main text or caption for readers. Additionally, is the Z-axis direction opposite? I think it should point into the page. According to the calculation, line 19 of the 7th part, we know the Δnpyro is 0.003, which is about 3 times larger than the required value of Δncrit of 0.00085. From my viewpoint, it seems that the pyroelectric effect is enough to explain the waveguide disappearance. This should be clarified if I misunderstood because other readers might think it in the same way. There are some typos, such as the word “coll” should be “cool” in line 3 of section 5.1, the word “piroelectric” should be “pyroelectric” in line 23 of the 7th part, the word “and” should be removed in line 32 of section 5.5, the word “by” should not be repeated in line 33 of the 6th The full text should be checked again carefully.
